# Cohort profile: population-based cohorts of patients with colorectal cancer and of their relatives in Geneva, Switzerland

Simone Benhamou ![ORCID] ,[1,2] Evelyne Fournier,[1] Giacomo Puppa,[3] Thomas McKee,[3] Frédéric Ris,[4] Laura Rubbia-Brandt,[3] Valeria Viassolo,[5] Thomas Zilli,[6,7] Inti Zlobec,[8] Pierre Olivier Chappuis,[5,9] Elisabetta Rapiti[1]

For numbered affiliations see end of article.

**Correspondence to**
Professor Simone Benhamou; simone.benhamou@inserm.fr

## ABSTRACT

**Purpose** Colorectal cancer (CRC) is the third leading cause of cancer death worldwide. Variability between patients in prognosis and treatment response is partially explained by traditional clinicopathological factors. We established a large population-based cohort of patients with CRC and their first-degree and second-degree relatives registered in the Canton of Geneva, to evaluate the role of family history and tumour biomarkers on patient outcomes.

**Participants** The cohort includes all patients with CRC diagnosed between 1985 and 2013. Detailed information on patient and tumour characteristics, treatment and outcomes were extracted from the Geneva Cancer Registry database, completed by medical records review and linkage with administrative and oncogenetics databases. Next-generation tissue microarrays were constructed from tissue samples of the primary tumour. A prospective follow-up of the cohort is realised annually to collect data on outcomes. First-degree and second-degree relatives of patients are identified through linkage with the Cantonal Population Office database and information about cancer among relatives is retrieved from the Geneva Cancer Registry database. The cohort of relatives is updated annually.

**Findings to date** The cohort includes 5499 patients (4244 patients with colon cancer and 1255 patients with rectal cancer). The great majority of patients were diagnosed because of occurrence of symptoms and almost half of the cases were diagnosed with an advanced disease. At the end of 2019, 337 local recurrences, 1143 distant recurrences and 4035 deaths were reported. At the same date, the cohort of first-degree relatives included 344 fathers, 538 mothers, 3485 children and 375 siblings. Among them, we identified 28 fathers, 31 mothers, 18 siblings and 53 children who had a diagnosis of CRC.

**Future plans** The cohort will be used for long-term studies of CRC epidemiology with focus on clinicopathological factors and molecular markers. These data will be correlated with the most up-to-date follow-up data.

## INTRODUCTION

In Europe, colorectal cancer (CRC) is the second most common malignancy in women and the third one in men.[1] In 2020, 281 714

## STRENGTHS AND LIMITATIONS OF THIS STUDY

⇒ This study, using a powerful resource based on regional record-linkage analysis of routinely collected data, will allow having one of the most comprehensive cohorts of patients with colorectal cancer (CRC), and their relatives, at a population level.

⇒ Considerable insights into the natural history of CRC and the mechanisms through which epidemiological, clinicopathological and molecular markers may alter the prognosis are expected.

⇒ Results of the study will provide evidence to guide health professionals in the stratification and management of patients with this disease, and surveillance of their relatives.

⇒ A weakness of the cohort is the absence of tumour samples for approximately 30% of patients with CRC.

⇒ Another limitation is the large amount of missing data for some of the recorded variables that will be addressed using multiple imputation procedures.

men and 238 106 women were diagnosed with CRC, and it accounted for approximately 13% of all cancer-related deaths in both genders. More than 80% of all CRC occur at the age of 60 years or more.

In high-income countries, the risk of CRC has been ubiquitously associated with environmental factors, such as dietary patterns (diet rich in fat and meat and poor in unrefined cereals and fibre), obesity and sedentary lifestyle.[2]

Approximately 6% of patients with CRC carry germline mutations in genes responsible of inherited genetic predisposition to CRC.[3] Lynch syndrome is the most common hereditary CRC syndrome and has been implicated in 2%–5% of newly diagnosed CRC cases. It is characterised by an autosomal dominant mode of inheritance and by a highly increased risk of CRC and endometrial cancer, as well as to a lesser extent of some other cancers (eg, ovary, stomach, small

intestine, biliary and urinary tract).[4] Germline pathogenic variants in the main DNA mismatch repair (MMR) genes, *MLH1*, *MSH2*, *MSH6* and *PMS2*, explain the majority of Lynch syndrome cases. Most Lynch syndrome-associated CRC tumours harbour a phenotype of high DNA microsatellite instability (MSI).

Up to 30% of the patients with CRC report having one or more relatives also diagnosed with the disease but are not consistent with one of the known inherited syndromes. Family history (FH) of CRC is a well-known risk factor for the development of the tumour,[5–9] but its impact on clinicopathological characteristics of CRC in patients is not entirely understood.[10–13] Also, it is not entirely clear how FH may affect CRC survival as the few studies performed on this topic have inconsistent findings.[7 14–20]

CRC is a heterogeneous disease with a variable natural history according to tumour, node, metastases (TNM) stage. There is growing evidence that CRC should be subdivided into different prognostic groups defined by robust molecular biomarker combinations and intratumour features of immune infiltration.[21] Such markers could guide the use of more or less aggressive treatment regimens and enable clinicians to balance expected outcomes against early and late therapeutic toxicities. This is particularly important in the case of locally advanced CRC (stages II and III), in which patients may be cured by surgery alone and only a portion derive benefit from adjuvant therapy. However, most molecular markers and immune-based studies have limited power, analyse small numbers of tumours (<500 in most studies), and only a few are population-based.[22–24] To date, although several molecular markers of outcome in CRC have been proposed, MSI status and mutations in oncogenes such as *KRAS* and *BRAF* are those most consistently used in clinical practice. However, patient outcomes remain variable, and stratification according to prognosis needs to be improved.

We set up a population-based cohort of patients with CRC and their first-degree and second-degree relatives (FDRs and SDRs, respectively) registered in the Canton of Geneva, to evaluate the role of FH and tumour biomarkers on patient outcome (locoregional and distant recurrences, survival) considering relevant clinicopathological features and treatments.

## COHORTS DESCRIPTION
### Cohort of patients with CRC
The cohort of patients with CRC has been established using the Geneva Cancer Registry (GCR) database which includes all incident cases of malignant neoplasms occurring since 1970 in the population of this Swiss canton (approximately 500 000 inhabitants).

Patients who were diagnosed with a primary invasive cancer of the colon, except appendix (International Classification of Diseases for Oncology ICD-O, C18.0, C18.2–C18.9), the rectosigmoid junction (ICD-O, C19) and the rectum (ICD-O, C20) between 1985 and 2013,

were eligible for inclusion in the CRC cohort. For colon cancer, the splenic flexure was considered as the frontier to define right-sided and left-sided tumours. Thus, right colon cancers included tumours occurring in the cecum, ascending colon, hepatic flexure and transverse colon (ICD-O codes: C18.0, C18.2–18.4). Left-sided tumours included tumours located at the splenic flexure, descending colon, sigmoid and rectosigmoid junction (ICD-O codes: C18.5–18.7, 19.9).

A prospective open-ended follow-up of all cancer cases registered in the GCR database since 1970 is realised annually to collect data on recurrences (locoregional and distant), second cancer and survival. These ongoing follow-up data are registered in the GCR database. For the present report, we used data on events that occurred up to 31 December 2019 in the CRC cohort.

Overall, 5985 CRC cases were registered in the GCR database between 1985 and 2013. Of these, 486 (8.1%) cases were subsequently excluded (mainly cases discovered from the death certificate or at the autopsy (n=217), cases with in situ cancer, cancer of other sites, Bauhin valve cancer, lymphomas, sarcomas, melanomas or no cancer (n=243), and lost medical charts (n=26)). Accordingly, the final cohort population includes 5499 patients (4244 patients with colon cancer (CC) and 1255 patients with rectal cancer (RC).

### Cohort of relatives of patients with CRC
Parents and offspring of patients with CRC have been identified through a computerised search in the administrative files of the Cantonal Population Office (CPO). The CPO maintains a register in which names, date of birth and personal CPO identification number are recorded for all residents in the Canton of Geneva since 1985. Using the CPO identification number of the patient with CRC included in the GCR data set, we can identify the name, identification number, date of birth and date of death or emigration (if applicable) of all first-degree 'vertical' links, that is, both the parents and the offspring. To trace siblings of the patients with CRC, the search is done using as key the CPO identification number of the parents identified in the previous step. A similar linkage process allows identifying SDR (ie, grandparents, aunts, uncles, grandchildren, nieces and nephews, half-brothers and half-sisters). An automated procedure to identify SDRs is being developed. The cohort of relatives will be updated at the end of every year using the annually updated GCR and CPO files.

Sources of data and links between the data sets are shown in figure 1.

## DATA COLLECTION
### Cohort of patients with CRC
Information collected for the 5499 patients with CRC included in the cohort is summarised in table 1.

Data on patient and tumour characteristics, treatment and outcomes were extracted from the GCR database.

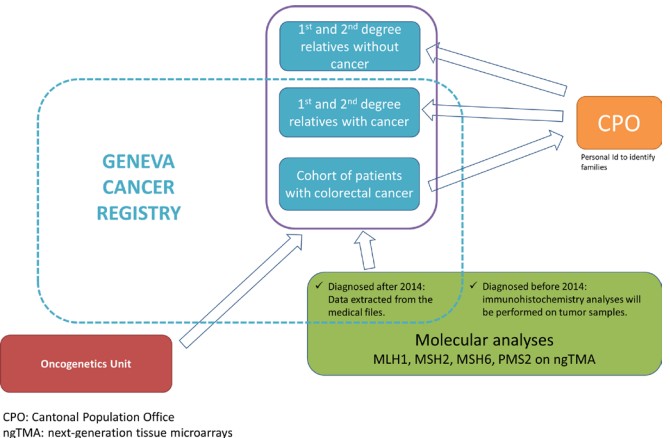

CPO: Cantonal Population Office
ngTMA: next-generation tissue microarrays

**Figure 1** Sources of data and links between the data sets.

The registry systematically records socio-demographic information (date of birth, gender, nationality, place of birth, marital status and occupation), diagnostic circumstances (origin of diagnosis, presence of symptoms and methods of assessment), tumour characteristics (histological type, differentiation coded according to ICD-O, stage of the disease at diagnosis (size of the tumour, clinical TNM, pathological TNM, Surveillance, Epidemiology and End Results, number of lymph nodes resected, number of positive lymph nodes), treatment during the first 6 months after diagnosis (type of surgery, radiotherapy, chemotherapy, etc), finality of the treatment (curative, palliative or not specified), surgical margin of the tumour resection (positive, negative), occurrence of recurrences (locoregional and distant), second tumour occurrence and, if applicable, date and cause of death.

**Table 1** Summary of data collected for patients with colorectal cancer and data source

| **Data collected routinely by the Geneva Cancer Registry** | |
|---|---|
| **Baseline** | |
| Demographics/socioeconomics | Date and place of birth, gender, marital status, nationality, last occupation |
| Medical history of cancer | Dates, sites, tumour characteristics, treatments |
| Diagnostic circumstances | Method of detection, presence of symptoms, methods of assessment |
| Treatments (first 6 months after diagnosis) | Type of surgery, radiotherapy, chemotherapy, finality (curative or palliative), surgical margins |
| Sector of care | Public or private |
| Tumour characteristics | Histological type and subtype, differentiation, TNM stage |
| **Follow-up** | |
| Locoregional and distant recurrences | Dates, sites, tumour characteristics, treatments |
| Second cancer | Dates, sites, tumour characteristics, treatments |
| Death | Date, cause and place |
| **Data collected from medical files** | |
| **Baseline** | |
| Lifestyle | History of tobacco consumption, alcohol habits |
| Anthropometrics | Weight, height |
| Medical history | Comorbidities, previous gastrointestinal diseases |
| Medication use | Aspirin, metformin, statins, vitamin D |
| Family history | History of cancer among FDRs and SDRs |
| Procedures prior diagnostic | Faecal occult blood tests, endoscopies, imaging |
| Origin of diagnosis | Colonoscopy, rectosigmoidoscopy, barium enema |
| Type of surgery | Elective or on emergency, presence of perforation |
| Tumour characteristics | Location, development of polyps, Dukes stage, lymphovascular invasion, necrosis |
| **Data collected from linkage with the administrative CPO database** | |
| FDRs and SDRs | Gender, dates of birth, dates of death |
| **Data collected from linkage with the oncogenetics unit database** | |
| Genetic counselling and germline testing | Uptake and results |

FDRs, first-degree relatives; SDRs, second-degree relatives; TNM, tumour, node, metastases.

**Table 2** Patient characteristics at first CRC diagnosis

| Patient characteristics | Colon cancer, n=4244 | Rectal cancer, n=1255 | Total, n=5499 |
|---|---|---|---|
| Gender, n (%) | | | |
| Male | 2116 (49.9) | 723 (57.6) | 2839 (51.6) |
| Female | 2128 (50.1) | 532 (42.4) | 2660 (48.4) |
| Age at diagnosis (years), n (%) | | | |
| <50 | 234 (5.5) | 96 (7.6) | 330 (6.0) |
| 50–59 | 568 (13.4) | 221 (17.6) | 789 (14.3) |
| 60–69 | 948 (22.3) | 346 (27.6) | 1294 (23.5) |
| 70–79 | 1275 (30.0) | 336 (26.8) | 1611 (29.3) |
| ≥80 | 1219 (28.7) | 256 (20.4) | 1475 (26.8) |
| Nationality, n (%) | | | |
| Swiss | 3241 (76.4) | 909 (72.4) | 4150 (75.5) |
| Other | 1003 (23.6) | 346 (27.6) | 1349 (24.5) |
| Marital status, n (%) | | | |
| Married/partnership | 2289 (53.9) | 724 (57.7) | 3013 (54.8) |
| Single/widowed/divorced | 1955 (46.1) | 531 (42.3) | 2486 (45.2) |
| Socioeconomic status, n (%) | | | |
| High | 670 (15.8) | 185 (14.7) | 855 (15.5) |
| Medium | 1679 (39.6) | 515 (41.0) | 2194 (39.9) |
| Low | 1056 (24.9) | 369 (29.4) | 1425 (25.9) |
| Unclassifiable/missing | 839 (19.8) | 186 (14.8) | 1025 (18.6) |
| Period of diagnosis, n (%) | | | |
| 1985–1989 | 666 (15.7) | 180 (14.3) | 846 (15.4) |
| 1990–1994 | 678 (16.0) | 216 (17.2) | 894 (16.3) |
| 1995–1999 | 722 (17.0) | 221 (17.6) | 943 (17.1) |
| 2000–2004 | 735 (17.3) | 207 (16.5) | 942 (17.1) |
| 2005–2009 | 803 (18.9) | 235 (18.7) | 1038 (18.9) |
| 2010–2013 | 640 (15.1) | 196 (15.6) | 836 (15.2) |
| Charlson Comorbidity Index, n (%) | | | |
| 0 | 1356 (32.0) | 538 (42.9) | 1894 (34.4) |
| 1–2 | 982 (23.1) | 279 (22.2) | 1261 (22.9) |
| 3–4 | 237 (5.6) | 66 (5.3) | 303 (5.5) |
| ≥5 | 474 (11.2) | 106 (8.4) | 580 (10.5) |
| Missing | 1195 (28.2) | 266 (21.2) | 1461 (26.6) |
| Familial history of CRC, n (%) | | | |
| No | 1195 (28.2) | 521 (41.5) | 1716 (31.2) |
| Yes | 249 (5.9) | 115 (9.2) | 364 (6.6) |
| Missing | 2800 (66.0) | 619 (49.3) | 3419 (62.2) |
| At least one genetic counselling uptake, n (%) | | | |
| No | 4160 (98.0) | 1228 (97.8) | 5388 (98.0) |
| Yes | 84 (2.0) | 27 (2.2) | 111 (2.0) |
| History of CRC in FDRs, n (%) | | | |
| No | 1195 (28.2) | 521 (41.5) | 1716 (31.2) |
| Yes | 249 (5.9) | 115 (9.2) | 364 (6.6) |
| Missing | 2800 (66.0) | 619 (49.3) | 3419 (62.2) |
| Colon cancer location | | | |

Continued

**Table 2** Continued

| Patient characteristics | Colon cancer, n=4244 | Rectal cancer, n=1255 | Total, n=5499 |
|---|---|---|---|
| Right colon | 1731 (40.8) | | |
| Left colon | 1964 (46.3) | | |
| Not specified/overlap | 549 (12.9) | | |

CRC, colorectal cancer; FDRs, first-degree relatives.

Additional clinical information including type and date of surgery (elective or on emergency), presence of perforation, presence of polyps and presence of comorbidities was collected in the clinical files of patients with CRC. A structured questionnaire was developed for the study and data were entered online by three registrars into a central database specifically designed to store all data.

Data on genetic counselling uptake and germline testing was also collected using a linkage procedure with the database established by the Oncogenetics Unit at University Hospitals of Geneva, the only centre providing genetic counselling for familial aggregation or CRC predisposition syndromes in the Geneva area since 1994.

### Cohort of relatives of patients with CRC

Information about cancer among relatives of patients with CRC will be retrieved from the GCR data set using their GCR identification number. All cancer diagnoses registered in the GCR since its creation in 1970 will be recorded.

Data on cancer occurrence in the families will be updated at the end of every year using the annually updated files of CPO and GCR.

### Patient and public involvement

Patients and/or the public were not involved in the design and conduct of this research. The Steering Committee of the cohort is currently in the process of defining the procedures to access the resources of the cohorts (data and/or biological samples) by public entities for conducting specific research projects. One patient with CRC will be included in an extended Committee. All regulatory procedures will be outlined in a charter that we will make available to project leaders.

### SAMPLE COLLECTION AND NEXT-GENERATION TISSUE MICROARRAY CONSTRUCTION

For each case, formalin-fixed paraffin-embedded (FFPE) tissue samples of the primary tumour and slides were searched in the pathology laboratories using biopsy/surgery report numbers extracted from pathology reports. FFPE blocks (excluding biopsies) were then collected and tissue re-embedded in fresh paraffin blocks if necessary. From each block, a new section was cut, H&E stained and scanned at high resolution on a whole slide scanner (Pannoramic 250 Flash II, 3DHistech). For next-generation tissue microarray (ngTMA) construction, four circular regions with 600 µm diameter were defined on the whole slide image, two for tumour centre and for two tumour border. Based on the coordinates defined by these regions, an ngTMA was constructed using a TMA Grand Master (3DHistech).

TMAs have been constructed for 3723 (67.7%) of the 5499 patients with CRC. The tumours are being currently classified as microsatellite stable or unstable on the basis of the expression of the main four DNA MMR proteins (MLH1, MSH2, MSH6, PMS2). Since 2014 these markers are analysed and reported routinely for each patient diagnosed with CRC. Immunohistochemical loss of expression of MLH1, MSH2, MSH6 and/or PMS2 expression will be used to suspect Lynch syndrome. Additional analysis and information (*BRAF* mutation status, methylation of *MLH1* promoter, FH, germline testing) will be necessary to confirm or exclude this predisposition syndrome to hereditary cancer. Other biomarkers, such as the *BRAF* V600E mutation status and the Immunoscore[25] will be assessed. We expect to correlate data on microsatellite status with updated follow-up data in 2021 and those on BRAF mutation and Immunoscore with updated follow-up data in 2022.

### FINDINGS TO DATE

Patient characteristics are shown in table 2.

The median age at diagnosis is 73 years (range: 16–101) for patients with CC and 68 years (range: 24–101) for patients with RC. Approximately 50% of cases with CC and 58% of cases with RC are men. Among the 4244 patients with CC, 41% had right CC, 46% left CC and 13% overlapping tumours or unknown CC location. Tumour and treatment characteristics of patients with CRC are shown in table 3.

Only 3% of the patients were diagnosed during a screening procedure. This low proportion is partly explained by the absence of a population screening programme in Geneva and lack of reimbursement of screening examinations before 2014. Thus, the great majority of patients with CC or RC were diagnosed because of occurrence of symptoms (80.6% and 86.3%, respectively). Abdominal emergency at diagnosis was fourfold more frequent for patients with CC than patients with RC (18% vs 5%). Cancer diagnosis was histologically confirmed in 90% of cases. About half of the patients with CC or RC were diagnosed with an advanced disease, that is, stages III and IV (table 3). Tumours were surgically resected for 82% of cases with

**Table 3** Tumour diagnosis and treatment characteristics at colorectal cancer diagnosis

| Tumour diagnosis/treatment | Colon cancer, n=4244 | Rectal cancer, n=1255 | Total, n=5499 |
|---|---|---|---|
| Reason leading to diagnosis, n (%) | | | |
| Population screening | 136 (3.2) | 37 (2.9) | 173 (3.1) |
| High risk | 39 (0.9) | 16 (1.3) | 55 (1.0) |
| Symptoms | 3421 (80.6) | 1083 (86.3) | 4504 (81.9) |
| Fortuitous discovery | 434 (10.2) | 72 (5.7) | 506 (9.2) |
| Others | 18 (0.4) | 6 (0.5) | 24 (0.4) |
| Missing | 196 (4.6) | 41 (3.3) | 237 (4.3) |
| Abdominal emergency, n (%) | | | |
| No | 2466 (58.1) | 976 (77.8) | 3442 (62.6) |
| Yes | 763 (18.0) | 63 (5.0) | 826 (15.0) |
| Missing | 1015 (23.9) | 216 (17.2) | 1231 (22.4) |
| Imaging use for diagnosis, n (%) | | | |
| PET scan | 16 (0.4) | 4 (0.3) | 20 (0.4) |
| CT scan/MRI | 780 (18.4) | 250 (19.9) | 1030 (18.7) |
| Others | 443 (10.4) | 113 (9.0) | 556 (10.1) |
| Stage, n (%) | | | |
| I | 657 (15.5) | 213 (17.0) | 870 (15.8) |
| II | 1262 (29.7) | 275 (21.9) | 1537 (28.0) |
| III | 950 (22.4) | 436 (34.7) | 1385 (25.2) |
| IV | 976 (23.0) | 205 (16.3) | 1182 (21.5) |
| Missing | 399 (9.4) | 126 (10.0) | 525 (9.5) |
| Differentiation, n (%) | | | |
| Well differentiated | 792 (18.7) | 243 (19.4) | 1035 (18.8) |
| Moderately differentiated | 2264 (53.3) | 681 (54.3) | 2945 (53.6) |
| Poorly differentiated | 563 (13.2) | 111 (8.9) | 674 (12.2) |
| Missing | 625 (14.7) | 220 (17.5) | 845 (15.4) |
| Blood vessels invasion, n (%) | | | |
| No | 2132 (50.2) | 430 (34.3) | 2562 (46.6) |
| Yes | 886 (20.9) | 186 (14.8) | 1072 (19.5) |
| Missing | 1226 (28.9) | 639 (50.9) | 1865 (33.9) |
| Lymph node invasion, n (%) | | | |
| No | 2159 (50.9) | 459 (36.6) | 2618 (47.6) |
| Yes | 612 (14.4) | 89 (7.1) | 701 (12.7) |
| Missing | 1473 (34.7) | 707 (56.3) | 2180 (39.6) |
| Surgery, n (%) | | | |
| No | 521 (12.3) | 207 (16.5) | 728 (13.2) |
| Yes | 3698 (87.1) | 1040 (82.9) | 4738 (86.2) |
| Missing | 25 (0.6) | 8 (0.6) | 33 (0.6) |
| Chemotherapy, n (%) | | | |
| No | 3146 (74.1) | 694 (55.3) | 3840 (69.8) |
| Yes | 934 (22.0) | 518 (41.3) | 1452 (26.4) |
| Missing | 164 (3.9) | 43 (3.4) | 207 (3.8) |
| Radiotherapy, n (%) | | | |
| No | 4131 (97.3) | 629 (50.1) | 4760 (88.6) |
| Yes | 97 (2.3) | 597 (47.6) | 694 (12.6) |
| Missing | 16 (0.4) | 29 (2.3) | 45 (0.8) |

PET, positron emission tomography.

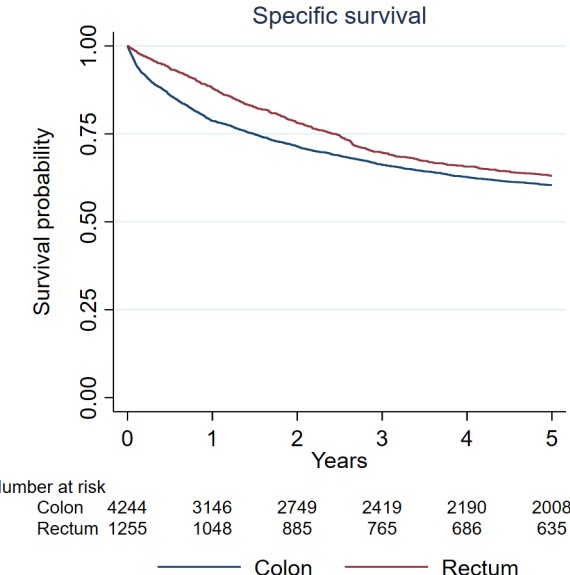

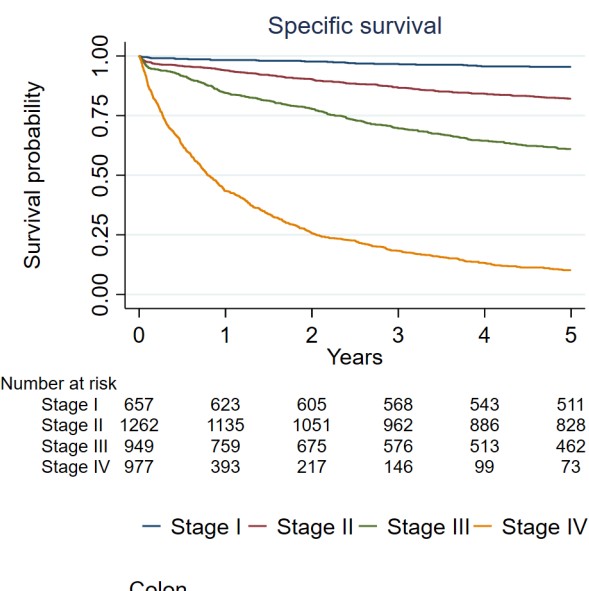

**Figure 2** Specific survival rates for colon cancer and rectal cancer.

**Figure 3** Specific survival rates for colon cancer according to stages.

CC and 87% of cases with RC. Patients with RC were more often treated with chemotherapy and radiotherapy than patients with CC.

At the end of 2019, 337 local recurrences, 1143 distant recurrences and 4035 deaths were reported. The median follow-up for the cohort was 4.5 years (Q1; 1.1 to Q3; 10.5). Five per cent of the patients were lost during follow-up. Patients with CRC were followed for survival from the date of diagnosis to the date of death or emigration or end of 2019, whichever came first. We considered both the CRC-specific survival time, defined as the interval between the date of diagnosis and the date of death from CRC, and the overall survival time, defined as the interval between the date of diagnosis and the date of death from any cause. Specific-survival rates for CCs and RCs and overall survival rates were calculated using the Kaplan-Meier method. The 5-year disease-free survival rate was 48% (95% CI: 47% to 50%) for CC and 51% (95% CI: 49% to 54%) for patients with RC. The 5-year overall survival rate was 49% (95% CI: 48% to 51%) for patients with CC and 53% (95% CI: 50% to 56%) for patients with RC. The 5-year cancer-specific survival rate was 61% (95% CI: 59% to 62%) for patients with CC and 63% (95% CI: 61% to 66%) for patients with RC (figure 2). Specific-survival rates were significantly different (logrank test, p<0.001) between stages among patients with CC (figure 3) and among patients with RC (figure 4).

At the end of 2019, the cohort of FDRs included 344 fathers, 538 mothers, 3485 children and 375 siblings of the 5499 patients with CRC. Among these family members, we identified 28 fathers, 31 mothers, 18 siblings and 53 children who had a diagnosis of CRC.

### Strengths and limitations

This study, using a powerful resource based on regional record-linkage analysis of routinely collected data will allow having, for each of the patient with CRC, the completeness of his/her FDR and SDR (identified by linkage with administrative population database) and the opportunity to identify in this cohort of relatives all those diagnosed with cancer by linkage with cancer registry data). Considerable insights into the natural history of CRC and the mechanisms through which epidemiological, clinicopathological and molecular markers may

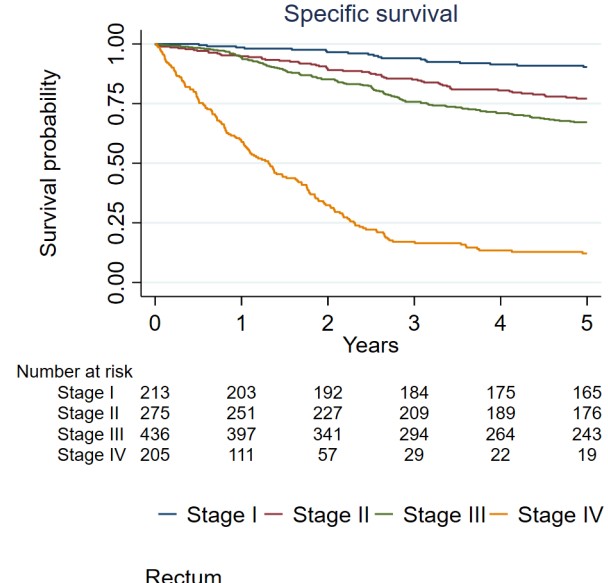

**Figure 4** Specific survival rates for rectal cancer according to stages.

alter the prognosis are expected. Results of the study will provide evidence to guide health professionals in the stratification and management of patients with this disease, and surveillance of their relatives.

Weaknesses of the cohort are the large amount of missing data for some of the recorded variables and the absence of tumour samples for approximately 30% of patients with CRC. We will use multiple imputation procedures to address the issue of missing data.

## COLLABORATION

The GCR welcomes collaboration with national and international research groups to evaluate the role of FH and tumour biomarkers on patient outcome (locoregional and distant recurrences, survival) considering relevant clinicopathological features and treatments.

The Steering Committee of the cohort is currently in the process of defining the procedures to access the resources of the cohorts (data and/or biological samples) by public entities for conducting specific research projects. One patient with CRC will be included in an extended Committee. All regulatory procedures will be outlined in a charter that we will make available on request to project leaders.

**Author affiliations**
¹Geneva Cancer Registry, Global Health Institute, University of Geneva, Geneva, Switzerland
²INSERM Unit 1018, Research Centre on Epidemiology and Population Health, Villejuif, Île-de-France, France
³Department of Clinical Pathology, Geneva University Hospitals, Geneva, Switzerland
⁴Division of Digestive Surgery, Geneva University Hospitals, Geneva, Switzerland
⁵Oncogenetics Unit, Division of Oncology, Geneva University Hospitals, Geneva, Switzerland
⁶Radiation Oncology, Geneva University Hospitals, Geneva, Switzerland
⁷Faculty of Medicine, University of Geneva, Geneva, Switzerland
⁸Institute of Pathology, University of Bern, Bern, Switzerland
⁹Division of Genetic Medicine, Geneva University Hospitals, Geneva, Switzerland

**Contributors** SB, GP, FR, LR-B, VV, TZ, IZ, POC and ER contributed the original concept and design of the study. EF, GP, LR-B, TM, EF and IZ supervised the data collection and sample management. EF developed the linkage procedures and conducted the statistical analyses. SB drafted the manuscript. All authors provided input and feedback on the paper drafts and approved the final version of the manuscript. SB is acting as the guarantor.

**Funding** The study was supported by grants from the Swiss Science National Foundation (320030-163342/1), the Swiss Cancer League (KFS-3932-08-2016-R and KFS-5243-02-2021) and the Geneva Cancer League (subvention 1813).

**Competing interests** None declared.

**Patient and public involvement** Patients and/or the public were not involved in the design, or conduct, or reporting, or dissemination plans of this research.

**Patient consent for publication** Not applicable.

**Ethics approval** Ethical approval for this study was obtained from Geneva Ethics Committee (Commission cantonale de la recherche scientifique de Genève (CCER), N° 2016-00141. Participants gave informed consent to participate in the study before taking part.

**Provenance and peer review** Not commissioned; externally peer reviewed.

**Data availability statement** Data are available upon reasonable request. Researchers may request access to data by contacting the corresponding author.

**ORCID iD**
Simone Benhamou http://orcid.org/0000-0003-1162-9165

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
