## [Reviewer comments · BMJ Open]

ARTICLE DETAILS

TITLE (PROVISIONAL)	Cohort profile: population-based cohorts of patients with colorectal cancer and of their relatives in Geneva, Switzerland
AUTHORS	Benhamou, Simone; Fournier, Evelyne; Puppa, Giacomo; McKee, Thomas; Ris, Frédéric; Rubbia-Brandt, Laura; Viassolo, Valeria; Zilli, Thomas; Zlobec, Inti; Chappuis, Pierre; Rapiti, Elisabetta

VERSION 1 – REVIEW

REVIEWER	Downing, A University of Leeds, Leeds Institute of Cancer and Pathology
REVIEW RETURNED	24-May-2022

GENERAL COMMENTS	This is a well-written manuscript describing a population-based cohort of colorectal cancer patients in Geneva, Switzerland. I have a few comments to improve the clarity of the information presented. Abstract & Strengths First and second-degree relatives are not mentioned in the abstract. This is a key strength and should be mentioned. In the Strengths it states this will allow having one of the most comprehensive cohorts of CRC patients and their relatives, at a population level. There are quite a few cancer cohorts/data repositories now, is this really one of the most comprehensive? It is relatively small and suffers from substantial missing data. Patient and public involvement Patients and/or the public were not involved in the design, conduct, reporting or dissemination of this research – why not? I'm sure that patients would like to be consulted. Although the registry has collected these data for many years, this is a new linkage and use of the data and it would be good to get their views. Patients generally like their data to be used and are supportive. Do you have a patient-public group that you could take this to? If other research groups request to use the data it might be good to involve patient(s) in the decision as to whether the project is given access to the data. Data Would be useful to have a diagram that details the different data sources and how they link together.
--

	Make it clearer in the Sample collection section that you will be looking at Lynch syndrome – this is mentioned in the introduction but not explicitly in the data collection or findings to date. Findings to date Quite a lot of missing data – e.g. for comorbidity, socioeconomic status, family history. How might this be improved? Collaboration More detail needed here. The guidance suggests: Authors should include a section on what data will be available, to whom, how it can be accessed and what restrictions to reuse may apply. Please also state what kind of collaboration you are encouraging. What will the cohort be used for? What are the initial research questions? Who will use the data? Can anyone request the data or have to be in Switzerland, in EU? What are the plans for the future?
--	--

REVIEWER	Aragonés, Nuria Madrid Regional Health Authority , Public Health Division
REVIEW RETURNED	19-Jun-2022

GENERAL COMMENTS	This is a cohort profile study describing a population based cohort of colorectal cancer patients and their relatives in Geneva, Switzerland. The protocol is clear and well written. The paper presents preliminary analyses of the colorectal cancer cohort, but lacks information on the status of the cohort of relatives, what limits the significance of the work in its present version. I also have some minor comments: 1) Why is the title about “the cohort of colorectal cancer patients” while the objective of the paper focussing on both the cohort of patients and the cohort of their first and second-degree relatives? 2) Is it possible to provide some basic details on the status of the cohort of relatives? The paper lacks information of the number and characteristics of relatives the authors have identified. I think this paper will improve when the data on the cohort of relatives are presented. 3) I suggest the authors include a discussion in which they would be able to deepen into the strengths and limitations of the cohort. 4) Page 7. SEER should be written in full. 5) Page 7. Line 57. The authors say “Data on genetic counseling uptake and germline testing was also be collected”. Did you want to say “Data on genetic counseling uptake and germline testing will also be collected”?
--

VERSION 1 – AUTHOR RESPONSE

Reviewer 1

This is a well-written manuscript describing a population-based cohort of colorectal cancer patients in Geneva, Switzerland. I have a few comments to improve the clarity of the information presented.

Abstract & Strengths

First and second-degree relatives are not mentioned in the abstract. This is a key strength and should be mentioned.

We fully agree that the cohort of relatives of patients with CRC is a key strength of our study. Two sentences have been added to the Abstract (page 2, lines 13-16 and lines 21-24) and the Findings to date section (page 11, lines 17-20): “First- and second-degree relatives of patients with CRC are identified through linkage with the Cantonal Population Office database and information about cancer among relatives is retrieved from the Geneva Cancer Registry database. The cohort of relatives is updated annually” and “At the end of 2019, the cohort of first-degree relatives included 344 fathers, 538 mothers, 3485 children, and 375 siblings of the 5499 patients with CRC. Among these family members, we identified 28 fathers, 31 mothers, 18 siblings and 53 children who had a diagnosis of colorectal cancer.”

In the Strengths it states this will allow having one of the most comprehensive cohorts of CRC patients and their relatives, at a population level. There are quite a few cancer cohorts/data repositories now, is this really one of the most comprehensive? It is relatively small and suffers from substantial missing data.

We agree that the size of our cohort of patients with CRC is relatively small. However, only few existing cohorts, including more than 5000 CRC cases with available tumor samples, are population-based and allow having, for each CRC patient, the completeness of his/her FDRs and SDRs (identified by linkage with administrative population database) and the opportunity to identify among the relatives all those diagnosed with cancer (by linkage with cancer registry data). A sentence has been added (page 11, lines 23-27).

We also agree that there is a large amount of missing data for some variables. We will use multiple imputation procedures to address this issue. This information has been added to the section Strengths and limitations of this study (page 3, lines 13-14).

Patient and public involvement

Patients and/or the public were not involved in the design, conduct, reporting or dissemination of this research – why not? I’m sure that patients would like to be consulted. Although the registry has collected these data for many years, this is a new linkage and use of the data and it would be good to get their views. Patients generally like their data to be used and are supportive. Do you have a patient-public group that you could take this to? If other research groups request to use the data it might be good to involve patient(s) in the decision as to whether the project is given access to the data.

The Steering Committee of the cohort is currently in the process of defining the procedures to access the resources of the cohorts (data and/or biological samples) by public entities for conducting specific research projects. There is a structure at the Geneva University Hospitals (Patients Partners +3P) helping to establish relationships between health professionals and patients for clinical and research purposes. We will contact them to include a patient in an extended committee. All regulatory

procedures will be outlined in a charter that we will make available to project leaders. The sections “Patient and public involvement” and “Collaboration” have been modified to provide this information (page 7, lines 28-33 and page 12, lines 7-11).

Data

Would be useful to have a diagram that details the different data sources and how they link together. A figure showing the sources of data and links between data sets has been added (Figure 1).

Make it clearer in the Sample collection section that you will be looking at Lynch syndrome – this is mentioned in the introduction but not explicitly in the data collection or findings to date.

The following sentence has been added in the Sample collection section (page 8, lines 15-19): “Immunohistochemical loss of expression of MLH1, MSH2, MSH6 and/or PMS2 expression will be used to suspect Lynch syndrome. Additional analysis and information (BRAF mutation status, methylation of MLH1 promoter, family history, germline testing) will be necessary to confirm or exclude this predisposition syndrome to hereditary cancer”.

Findings to date

Quite a lot of missing data – e.g. for comorbidity, socioeconomic status, family history. How might this be improved?

We will use multiple imputation procedures to address the issue of the large amount of missing data for some of the recorded variables. This sentence has been added (page 3, lines 13-14 and page 11, lines 33-34).

Collaboration

More detail needed here. The guidance suggests: Authors should include a section on what data will be available, to whom, how it can be accessed and what restrictions to reuse may apply. Please also state what kind of collaboration you are encouraging.

What will the cohort be used for? What are the initial research questions?

Who will use the data? Can anyone request the data or have to be in Switzerland, in EU?

What are the plans for the future?

As requested, more details have been added in the Collaboration section (page 12, lines 3-11) as follows: “The Geneva Cancer Registry welcomes collaboration with national and international research groups to evaluate the role of FH and tumor biomarkers on patient outcome (loco-regional and distant recurrences, survival) considering relevant clinico-pathological features and treatments. The Steering Committee of the cohort is currently in the process of defining the procedures to access the resources of the cohorts (data and/or biological samples) by public entities for conducting specific research projects. One patient with colorectal cancer will be included in an extended Committee. All regulatory procedures will be outlined in a charter that we will make available upon request to project leaders.”

Reviewer 2

This is a cohort profile study describing a population-based cohort of colorectal cancer patients and their relatives in Geneva, Switzerland. The protocol is clear and well written. The paper presents preliminary analyses of the colorectal cancer cohort, but lacks information on the status of the cohort of relatives, what limits the significance of the work in its present version. I also have some minor comments:

1) Why is the title about “the cohort of colorectal cancer patients” while the objective of the paper focusing on both the cohort of patients and the cohort of their first and second-degree relatives? We completed agree. The title has been changed as follows: Population-based cohorts of patients with colorectal cancer and of their relatives in Geneva, Switzerland

2) Is it possible to provide some basic details on the status of the cohort of relatives? The paper lacks information of the number and characteristics of relatives the authors have identified. I think this paper will improve when the data on the cohort of relatives are presented.

Some basic results on the cohort of first-degree relatives have been added in the Abstract (page 2, lines 24-28) and Findings to date (page 11, lines 17-20): “At the end of 2019, the cohort of first-degree relatives included 344 fathers, 538 mothers, 3485 children, and 375 siblings of the 5499 patients with colorectal cancer. Among these family members, we identified 28 fathers, 31 mothers, 18 siblings and 53 children who had a diagnosis of colorectal cancer.”

3) I suggest the authors include a discussion in which they would be able to deepen into the strengths and limitations of the cohort.

The following paragraph has been added (page 11, lines 23-34): “This study, using a powerful resource based on regional record-linkage analysis of routinely collected data will allow having, for each of CRC patient, the completeness of his/her FDR and SDR (identified by linkage with administrative population database) and the opportunity to identify in this cohort of relatives all those diagnosed with cancer (by linkage with cancer registry data). Considerable insights into the natural history of CRC and the mechanisms through which epidemiological, clinico-pathological, and molecular markers may alter the prognosis are expected. Results of the study will provide evidence to guide health professionals in the stratification and management of patients with this disease, and surveillance of their relatives.

Weaknesses of the cohort are the large amount of missing data for some of the recorded variables and the absence of tumor samples for approximately 30% of patients with CRC. We will use multiple imputation procedures to address the issue of missing data.”

4) Page 7. SEER should be written in full.

SEER has been written in full (page 7, lines 4-5)

5) Page 7. Line 57. The authors say “Data on genetic counseling uptake and germline testing was also be collected”. Did you want to say “Data on genetic counseling uptake and germline testing will also be collected”?

We have modified the sentence as follows: “Data on genetic counseling uptake and germline testing was also collected” (page 7, line 15).

VERSION 2 – REVIEW

REVIEWER	Downing, A University of Leeds, Leeds Institute of Cancer and Pathology
REVIEW RETURNED	20-Jul-2022
GENERAL COMMENTS	The authors have responded to my comments. However, I cannot see the new figure 1 in the files.